# West Nile Virus Neuroinfection in Humans: Peripheral Biomarkers of Neuroinflammation and Neuronal Damage

**DOI:** 10.3390/v14040756

**Published:** 2022-04-04

**Authors:** Orianne Constant, Jonathan Barthelemy, Anna Nagy, Sara Salinas, Yannick Simonin

**Affiliations:** 1Pathogenesis and Control of Chronic and Emerging Infections, University of Montpellier, INSERM, EFS, 34000 Montpellier, France; orianne.constant@etu.umontpellier.fr (O.C.); jonathan.barthelemy@inserm.fr (J.B.); sara.salinas@inserm.fr (S.S.); 2National Reference Laboratory for Viral Zoonoses, National Public Health Center, 1097 Albert Flórián Road 2-6, 1097 Budapest, Hungary; nagy.anna@nnk.gov.hu

**Keywords:** West Nile virus, flavivirus, neuroinvasive disease, peripheral biomarkers

## Abstract

Among emerging arthropod-borne viruses (arbovirus), West Nile virus (WNV) is a flavivirus that can be associated with severe neuroinvasive infections in humans. In 2018, the European WNV epidemic resulted in over 2000 cases, representing the most important arboviral epidemic in the European continent. Characterization of inflammation and neuronal biomarkers released during WNV infection, especially in the context of neuronal impairments, could provide insight into the development of predictive tools that could be beneficial for patient outcomes. We first analyzed the inflammatory signature in the serum of WNV-infected mice and found increased concentrations of several inflammatory cytokines. We next analyzed serum and cerebrospinal-fluid (CSF) samples from a cohort of patients infected by WNV between 2018 and 2019 in Hungary to quantify a large panel of inflammatory cytokines and neurological factors. We found higher levels of inflammatory cytokines (e.g., IL4, IL6, and IL10) and neuronal factors (e.g., BDNF, GFAP, MIF, TDP-43) in the sera of WNV-infected patients with neuroinvasive disease. Furthermore, the serum inflammatory profile of these patients persisted for several weeks after initial infection, potentially leading to long-term sequelae and having a deleterious effect on brain neurovasculature. This work suggests that early signs of increased serum concentrations of inflammatory cytokines and neuronal factors could be a signature underlying the development of severe neurological impairments. Biomarkers could play an important role in patient monitoring to improve care and prevent undesirable outcomes.

## 1. Introduction

West Nile virus (WNV) is an arthropod-borne virus (arbovirus) belonging to the genus *Flavivirus* in the family *Flaviviridae* and is a member of the Japanese encephalitis virus (JEV) serocomplex [1]. WNV is maintained in the environment through an enzootic cycle involving mosquitoes (*Culex* spp.) as vectors and birds as primary reservoir hosts, while spread to mammals is occasional. WNV infection in humans results in different outcomes, ranging from asymptomatic to severe neuroinvasive disease. Approximately 80% of cases are asymptomatic while others lead primarily to WNV fever (WNF) with mild flu-like febrile illness whereas ≈1% of cases display a WNV neuroinvasive disease (WNND) [1,2,3]. The neuroinvasive disease typically manifests as meningitis, meningoencephalitis, encephalitis, or acute-flaccid paralysis, which are commonly associated with diarrhea/vomiting, weakness, vision impairment, confusion, or somnolence [4]. Thus, inter-individual responses to WNV infection are highly heterogeneous. WNV was first detected in the West Nile region of Uganda in 1937 from the blood of a febrile woman [5]. The geographical distribution of WNV has greatly expanded over the last two decades, leading to nearly global distribution. WNV is currently common and can cause outbreaks in Africa, the Middle East, India, Australia, Central and Southern Europe, and in North, Central and South America [6,7]. The largest European WNV epidemic was in 2018, with 181 recorded deaths and more human cases than in the previous seven years combined [8].

WNV is a spherical, enveloped virus approximately 50 nm in diameter. The genome is a positive single-stranded RNA molecule of approximately 11 kilobases in length encoding three structural and seven non-structural proteins [9]. WNV has great genetic diversity with at least eight different lineages, among which two major lineages (lineages 1 and 2) are responsible for the outbreaks observed in Europe over the past years. Acute WNV infection in humans induces the production of various cytokines, chemokines, and other factors that play an important role in antiviral immunity. On the other hand, the post-infection inflammatory response is thought to have a critical role in disease pathogenesis [10]. The innate immune response to WNV infection is mainly initiated by Toll-like receptors 7 and 3 (TLR7, TLR3). Following recognition of single-stranded viral RNA or double-stranded viral RNA (while genomic RNA is replicating), TLR7 and TLR3 activate IRF-3 and NF-κB signaling pathways, inducing the synthesis of type I IFN, as well as a variety of pro-inflammatory cytokines, including TNF-α, IL-6, IL-1β, and IL-12; these mediate early antiviral responses. This initial antiviral response limits viral dissemination and likely underlies the majority of asymptomatic cases [11]. Nonetheless, high-level viremia may occur in some patients, associated with major host immune responses which can participate in viral spread. This spread occurs notably into the central nervous system (CNS), giving WNV a role in severe diseases, such as neurological disorders [12,13]. WNV gains access to the CNS via multiple pathways, including infection of olfactory and peripheral neurons with direct retrograde axonal transport [14], infiltration of infected immune cells (known as the Trojan-horse mechanism) [15,16], direct infection of brain barriers [17], or transport across brain barriers facilitated by their disruption during host inflammation [18]. CNS infection also exacerbates the host immune response by secretion of cytokines, chemokines, and other neuro-inflammatory mediators into the serum and cerebrospinal fluid (CSF) [19]. Inversely, the circulation of pro-inflammatory mediators participates in the disruption of brain barrier permeability and facilitates greater propagation of WNV in the CNS [20]. Both inflammation and neuronal response are important parameters associated with the outcomes for WNV-infected patients. The evaluation of these factors in blood and CSF samples (biomarkers) can allow for patient monitoring and improve patient care [21,22].

To date, several studies and case reports have suggested long-term consequences of WNV infection, such as prolonged fatigue until six months after infection [23], WNV retinopathy [24], or prolonged neuromuscular symptoms [25]. Persistent neuroinflammation in WNND patients could also affect the development of neurodegenerative diseases [26,27]. The ability to predict clinical evolution to either recovery or CNS involvement has not been yet elucidated. Thus, the identification of biomarkers associated with disease severity could enable better forecasting of WNV disease progression (i.e., the distinction between WNND and WNF) and could improve our knowledge of WNV-caused brain alteration in light of defining novel potential drug targets. The main aim of this study was to assess the inflammatory profile following WNV infection. We examined the differences in levels of cytokines and chemokines in infected mice and in patients with WNND and WNF, notably in serum and CSF, in the context of clinically relevant parameters.

## 2. Materials and Methods

### 2.1. Cells and Virus Strains

C6/36 cells (ATCC CRL-1660) and Vero cells (ATCC CCL81) were respectively cultured in Roswell Park Memorial Institute culture medium (RPMI) or Dulbecco’s modified Eagle’s medium (DMEM) with 10% heat-inactivated fetal bovine serum, 100 U penicillin, 100 µg/mL streptomycin supplementation. C6/36 cells were maintained at 28 °C whereas Vero cells were maintained at 37 °C with 5% CO_2_. The WNV lineage 2 (WNV-3125/France/2018) strain was provided by ANSES (National Agency for Food, Environmental and Occupational Health Safety, 94700 Maisons-Alfort, France). It was amplified in a limit of four times on C6/36 cells and the viral stocks were prepared by infecting 70% confluent C6/36 cells. Supernatants were collected 5 to 7 days after infection. Viral titers were determined on Vero cells using the Spearman-Kärber method [28] and expressed as TCID50/mL.

### 2.2. Mouse Experiments

C57/BL6 mice of three-weeks-old were provided from Janvier Laboratories (France). Mice were bred and maintained in a biosafety level 3 (BSL-3) animal facility according to the French Ministry of Agriculture and European institutional guidelines (Appendix A STE n°123). They were inoculated by subcutaneous injection with 10^3^ TCID50/mice of WNV or with an equivalent volume amount of PBS for the mock-infected group (n = 9 per group). At the end of the experiments or if mice presented health deterioration, they were euthanized by cervical dislocation and sera were collected and frozen at −80 °C. According to European institutional guidelines (Appendix A STE n°123) and the French Ministry of Agriculture, mice were bred and maintained. Experiments were approved by the French ethics committee (approval N° 6773–201609161356607).

### 2.3. Patients and Sample Collection

A total of 183 patients from Hungary with WNV infection detected during two consecutive transmission seasons (2018–2019) were available for the study. In patients with WNND (n = 58), serum, and for some of them CSF, samples were analyzed. In patients with WNV fever (n = 26), serum samples were analyzed. WNV diagnosis was confirmed according to the EU case definition for diagnosing and reporting WNV infection by detection of (a) WNV RNA in blood or CSF, (b) WNV IgM antibodies in CSF, or (c) WNV IgM high titer in serum and detection of WNV IgG and confirmation by virus neutralization test (VNT). This cohort harbors 78 (42.6%) females (median age: 51 years; interquartile range (IQR): 44–72 years) and 105 (57.4%) males (median age: 63 years; IQR 54–71 years). Patients had WNF (n = 52; 44 years; IQR 35–52 years) or WNND (n = 131; 62 years; IQR 51–71 years). The main symptoms are fever (79.92%), exanthema (71.15%), and headache (32.69%) for WNF patients and for WNND patients: meningitis (37.40%), encephalitis (15.27%), meningoencephalitis (5.34%) and encephalomyelitis (2.29%) associated with fever (67.94%), headache (41.22%), confusion (28.24%), vomiting (10.92%), dizziness (6.52%), and unconsciousness (5.34%). A control population was used in this study with sera samples from outpatients attending hospitals for occupational medicine without detected disease or infection. The study was conducted according to the guidelines of the Declaration of Helsinki and approved by the Ethics Committee of the Hungarian National Public Health Center (protocol code 45554-1/2020/MRLAB, approved on 20 September 2020).

### 2.4. Multiplex Assay

The quantification of mouse and human inflammatory factors was performed using the ProcartaPlex assay and recorded with a Luminex apparatus (MAGPIX; Thermo Fisher Scientific, Waltham, MA, USA). For human samples, 20 inflammatory cytokines were quantified in sera and CSF using a ProcartaPlex Human Inflammation Panel 20plex (Thermo Fisher Scientific, Waltham, MA, USA) and 18 neurological inflammatory factors were quantified with a ProcartaPlex Human Neuroscience Panel 18-plex (Thermo Fisher Scientific, Waltham, MA, USA). For the study of the mouse inflammatory response, a ProcartaPlex Mouse Cytokine and Chemokine Convenience Panel 1A 36-plex (Thermo Fisher Scientific, Waltham, MA, USA) were used. Both were conducted according to the manufacturer’s procedure.

### 2.5. In Vitro Human Blood-Brain Barrier Model

Human umbilical cord bloods were collected after an infant’s parents signed a consent form in compliance with French legislation. From these and following a previously published protocol [29], CD34^+^ blood-derived endothelial cells were cultured on matrigel-coated transwell filters (Costar, 0.4 µm). These cells were placed on top of bovine brain pericytes for 5 to 7 days allowing differentiation to human brain-like endothelial cells (hBLECs) that achieved human blood-brain barrier (BBB) characteristics as tight-junction proteins and transporters expression [30,31]. During this period, the medium was changed every 2 days and before experiments, the endothelial permeability coefficient (Pe) was measured using Lucifer Yellow (LY) (Life Technologies, Van Allen Way Carlsbad, California, United States, 20 µM). Before and after experiments on the human BBB model, Pe was measured after detection of the paracellular passage of LY by fluorescence detection on a Tecan SPARK 10M machine (432/538 nm of excitation/emission wavelength settings). With Pe < 1 × 10^−3^ cm/min, the endothelium was considered impermeable and was considered to be disturbed with Pe > 1 × 10^−3^ cm/min [32]. To evaluate the effect of WNND-serum on the endothelium, 50 µL of healthy-control or WNND serum mixed in 250 µL of medium were added to hBLECs for 4 days. Then the Pe was calculated as described below.

### 2.6. Statistical Analyses

All analyses of the multiplex assay were conducted using Prism software (Graphpad Prism 8, 9.3. 1, Dr. Harvey Motulsk, University of California, San Diego, United States) using unpaired *t*-tests, Welch’s *t*-tests, or Mann–Whitney tests following the number of samples compared.

## 3. Results

### 3.1. Characterization of Systemic Inflammation in Immunocompetent Mice

Previous studies in immunocompetent mice have shown that both innate and adaptive immune responses participate to control WNV disease [33,34,35]. To evaluate the systemic inflammatory response following WNV infection, we first assessed inflammation in the serum of infected wild-type C57/BL6 mice (described to be sensitive to WNV infection). Tissue (particularly the brain) cytokine profiles were investigated in several studies, but the serum cytokine profile in mice models remains poorly described [36,37,38]. Mice were inoculated by subcutaneous (sc.) injection and sera were collected six days post-infection (dpi). To characterize the immune profiles associated with the acute phase of WNV infection, we used an enzyme-linked immunosorbent assay (ELISA) multiplex assay containing different sets of soluble markers (a total of 36 molecules). We considered these markers valuable for assessment based on their reports in the literature in clinical cases of flavivirus infection. We detected significantly higher concentrations of cytokines (IL6, IL12p70, IL22, IL31, TNFα) and chemokines (CCL2, CCL4, CCL7, CCL11, CXCL1, CXCL2, CXCL10) in infected mice compared to control mice, reflecting a strong systemic inflammation in WNV-infected mice (Figure 1).

### 3.2. WNV Infection Leads to Systemic Inflammation, Particularly in Patients with Neuroinvasive Disease

WNV is one of the most incident viral zoonotic infections in Hungary, which was exposed to its largest WNV outbreak in 2018 where 215 diagnosed human cases were recorded. To characterize the inflammatory profile of patients exposed to WNV infection, we analyzed sera collected between 2018 and 2019 from 183 WNV-infected patients from Hungary. Paired samples of serum and CSF samples were available and analyzed for 19 patients with WNND. We first analyzed the expression of 20 cytokine and chemokine mediators: innate immunity factors (such as IFNα, IL-1α, IL6, TNFα); several other pro- and anti-inflammatory cytokines (GM-CSF, IFNγ, IL-1β, IL-4, IL-8, IL-10, IL-12p70, IL-13, IL-17A (CTLA-8)), and chemokines (IP-10 (CXCL10), MCP-1 (CCL2), MIP-1α (CCL3), MIP-1β (CCL4)). Comparing sera from non-infected patients to WNV-infected patients showed a global upregulation of cytokines (mainly interleukins) and chemokines in infected patients compared to healthy controls (Figure A1). More interestingly, the systemic inflammatory profile was different between patients with WNF versus WNND (including patients with encephalitis, meningitis, or meningoencephalitis). We found significantly higher concentrations of pro-inflammatory markers in WNND patients. Overall, several inflammatory factors were found more elevated in the WNND group than in the WNF or control group; with the WNF group displaying no difference from the control group (Figure 2). Precisely, WNND patients demonstrated more increased concentrations of IL1β, IL1α, IL4, IL8, IL10, IL13, IL17, and CCL2. IFNα and IFNγ were also upregulated in the sera of the WNND group. Within the WNND group, we did not observe a significant difference in the concentration of pro-inflammatory cytokines between patients with meningitis, encephalitis, or meningoencephalitis. Several cytokine expressions did however appear slightly higher in patients with meningitis (Figure A2).

### 3.3. Identification of Neuroinflammation and Neuronal Damage Factors in WNND

We then questioned the presence of neuroinflammation and neuronal damage factors in the CSF of WNND patients presenting strong systemic inflammation. To investigate this, we used ELISA targeting 18 neuronal markers described as having a role in neuroinflammation and/or neurodegenerative diseases (such as Alzheimer’s disease (AD), amyotrophic lateral sclerosis (ALS), dementia, etc.). The markers included: β-amyloid (1–42), brain-derived neurotrophic factor (BDNF), YKL-40 (also known as Chitinase 3-like 1 (CHI3L1)), ciliary neurotrophic factor (CNTF), fibroblast growth factor-21 (FGF-21), glial cell line-derived neurotrophic factor (GDNF), glial fibrillary acidic protein (GFAP), Kallikrein-6 (KLK6), macrophage migration inhibitory factor (MIF), neuronal cellular adhesion molecule 1 (NCAM-1), Neurogranin (NRGN), Neurofilament H (NF-H), nerve growth factor β (β-NGF), S100 Calcium Binding Protein B (S100B), Tubulin associated unit (Tau (total)), Tau (pT181), TAR DNA-binding protein (TDP-43), and Ubiquitin C-terminal hydrolase L1 (UCHL1). Due to the limited availability of CSF samples from healthy individuals or WNF patients given the ethical implications, we did not have a control group for comparison of the results from CSF samples of WNND patients. We, therefore, decided to compare CSF samples from patients with meningitis versus those with encephalitis (meningitis and encephalitis have the highest incidence among our patients with WNND). (Figure 3A). We found that CSF from WNND patients contains a high concentration of neuronal biomarkers, such as amyloid-β, GFAP, Kallikrein-6, NF-H, S100N, Tau, TDP-43, and UCHL1 (Figure 3A).

It is not unexpected to find a high concentration of these markers in CSF. However, interestingly we also found a significant increase in some of these markers in the sera from patients with WNND, unlike WNF patients. MIF, NCAM1, TDP-43, and YKL-40 (CHI3L1) were significantly increased compared to healthy controls (found at equivalent levels in serum and CSF samples) (Figure 4). We also observed an increase in GFAP, GDNF, KLK6, and BDNF in the serum of patients with WNND, but to a lesser extent than in CSF samples. Overall, we did not find a significant difference for the majority of markers between patients with meningitis versus with encephalitis. Nevertheless, some markers differed in concentration in CSF samples, such as IFNγ and Neurogranin; these were significantly higher in patients with encephalitis. Conversely, GDNF was specifically higher in the CSF of patients with meningitis (median: 14 pg/mL; IQR 14–50.3 pg/mL) (Figure 3B).

### 3.4. Systemic Inflammation Is Persistent

Another important parameter during the course of WNV infection is the duration of inflammation [39]. To determine whether cytokine/chemokine induction differed according to time, we were able to assess inflammation in the sera of several infected patients over a short-term (0–11 days post-infection) and long-term (>20 days post-infection) period compared to healthy controls. We did not observe late inflammation for patients with WNF, and serum cytokine and chemokine levels remained low both in the short and long term (Figure 5). On the other hand, we observed the maintenance of a long-term inflammatory context for the majority of patients with WNND for many of the cytokines/chemokines studied (changes not always significant). Cytokine and chemokine levels tended to decrease over time but remained very high for certain proteins, such as IL8, IL13, and CCL2 (Figure 5). In some patients, we were even able to detect an increase in certain cytokines, such as CCL3 or CCL4, 33 days after the suspected date of infection (Figure A3). This long-term inflammation following WNV infection among patients developing neuronal disease can have very deleterious effects on health, including prolonged fatigue, irreversible neurological impairments, or neurocognitive alterations. Furthermore, other neuronal injuries can include vascular leakage by the direct effect of WNV or the effects of systemic inflammation on the BBB [40].

### 3.5. Serum from Patients with WNND Could Affect BBB Integrity

The increase in cytokine and chemokine levels after WNV infection could play a role in vascular endothelium leakage already described in many case reports [41]. In addition to the high-level and prolonged inflammation found in the sera of WNND patients, as well as the presence of neuronal markers, these patients are at risk of developing vascular leakage in the CNS. We, therefore, investigated the effect of the inflammatory environment on the cerebral endothelium integrity. We incubated sera from WNND patients or healthy controls using an innovative human in vitro BBB model recapitulating the main characteristics of the endothelial barrier observed in vivo [29,42,43,44,45,46]. Briefly, CD34^+^ cord blood-derived hematopoietic stem cell-derived endothelial cells were allowed to differentiate on culture inserts in contact with pericytes for 6–7 days to acquire BBB characteristics (Figure 6A). The permeability coefficient (Pe was measured four days post-incubation by measuring yellow Luciferase passage across the endothelium (medium required changing after to maintain cells in appropriate physiological conditions). Mock-infected endothelium (exposed to healthy control serum) demonstrated a Pe of ~0.62 × 10^−3^ cm/min, consistent with “tight” BBB endothelia. Serum from patients with WNND induced a slight increase in Pe (~0.69 × 10^−3^ cm/min). This value is still consistent with a “tight” BBB, suggesting that in our conditions WNV serum did not vastly impair BBB integrity, but may trigger a subtle effect on barrier integrity that could increase access of cytokines/chemokines to the brain or perturb vascular functions (Figure 6B).

## 4. Discussion

The clinical outcomes of WNV infection can vary from asymptomatic cases and mild symptoms (fever) to neurological impairments in the more severe forms. These outcomes depend on factors such as age or virus strain but also host genetics factors or coinfection with parasites may also dictate susceptibility to WNV infection [47,48,49]. A better knowledge of the inflammation associated with WNV infection is an important element for better understanding the underlying pathogenesis, and in particular the mechanisms involved in mild flu-like febrile illness or WNV-induced neurological disorders. Very few studies have evaluated the human immune response to WNV infection, and even more so in the context of serum and CSF samples from patients with WNV or WNND in a clinical setting [10,22,50,51,52]. In this study, we aimed to determine the mouse and human inflammatory profile following WNV infection, as well as biomarkers specific to the neuro-infection. Overall, we provide new insights on systemic inflammation following WNV infection. Firstly, using mouse models we found a strong systemic inflammatory response in the serum of WNV-infected mice with upregulation of a panel of cytokines and chemokines. The inflammatory profile observed in the serum of WNV-infected mice is similar to that of the brain inflammatory profile of WNV-infected mice. Both profiles are associated with elevated concentrations of a broad spectrum of cytokines or chemokines such as TNFα, IL6, CCL2, CCL4, and CCL7 [35,53,54].

When we compared serum from WNV-infected patients to healthy controls we also detected higher levels of inflammation: nine (including IL1α, IL4, IFNα) out of twenty markers were significantly increased. Specifically, patients developing WNND (pooling meningitis, meningoencephalitis, and encephalitis) have significantly increased levels of several pro-inflammatory mediators in comparison to healthy and WNF-infected individuals. This increased inflammation could be due to uncontrolled anti-viral responses and could participate in neuronal symptom development given that many brain infections lead to exacerbated immune response and associated-neuronal damage [52]. Indeed, a key feature during viral infections, whether they affect the brain or not, is the induction of inflammatory responses that can be both beneficial (control viral replication) or detrimental (trigger cellular toxicity). We observed the increased response of type 2 cytokines (IL4 and IL13) and IL10, an immunomodulatory interleukin, in WNND patients. Type 2 cytokines are crucial to the pathogenesis of many diseases. They suppress the development of protective type 1 immunity to a wide range of viral, bacterial, and protozoan pathogens and could contribute to regulating the excessive inflammatory response in WNND patients, in combination with IL10 [55]. In light of determining the molecular signature of WNV infection to better orientate patient care (i.e., anticipate severe symptoms), we also compared groups of patients with encephalitis, meningoencephalitis, and meningitis. We did not find significantly different levels of any relevant cytokines or chemokines in the sera to allow for distinction between these groups while there are only some differences in the CSF, including IFNγ, GDNF, or neurogranin. Patients who initially presented encephalitis have high rates of persistent neurological abnormalities several years after infection compared with those who presented meningitis [56]. It will be interesting to perform longer-term analyses to determine if certain of these biomarkers or other factors might be involved in this unfavorable evolution.

WNV infection can be deleterious to the brain as the CNS can undergo direct neuronal infection (WNV may infect neurons, microglia, and astrocytes), but also due to the indirect effects of global neuroinflammation and post-infectious mechanisms that can occur in distinct anatomical regions [44]. Severe and persistent neuroinflammation can trigger neurological disorders, including encephalitis, paralysis, ophthalmological impairments, or developmental defects, which in some cases can lead to long-term CNS defects. For example, the release of pro-inflammatory factors in the serum (IL6 or TNFα) can perturb brain barrier integrity and trigger neuronal impairments [57]. Direct WNV-CNS infection leads to neuroinflammation and neurotoxicity [58]. Prolonged systemic inflammation can activate pathophysiological axes (e.g., lipid and lipoprotein metabolism) and is related to a number of diseases (i.e., depression, other brain diseases, or persistent fatigue) [59,60,61]. In another aspect, control of early WNV infection steps is crucial; the initial type I IFN response must be controlled, but should also occur throughout the course of infection to avoid severe damage [22]. WNND patients in our study here showed high serum levels of inflammatory molecules for over 20 days, an effect that can cause many adverse effects. Neuroinvasion is a serious complication, with long-term sequelae that include ocular involvement, cognitive impairment, muscle weakness, and flaccid paralysis [62]. Previous studies have shown significantly higher concentrations of IFNα, IL-2, IL-6, IL-12p70, CXCL10, and GM-CSF in patients with a clinical diagnosis of prolonged post-infection fatigue (>6 months) reporting a history of symptomatic WNV infection [23]. More recently, another study reported elevated levels of TNF-α up to 36 months post-onset of illness in four patients with serologically-confirmed WNND. These patients had persistent post-infectious symptoms, suggesting a potential role for this cytokine in the extended post-inflammatory state and long-term morbidity associated with WNV infection [63].

Patients generally recover from WNF within a few days without deleterious effects, whereas WNND forms are more prone to the development of long-term impairments; hence the importance of patient follow-up during infection evolution, notably by the assessment of biomarkers that can orient care and anticipate neurological outcomes. Indeed, biomarkers are under intensive investigation as potential predictors of neurological impairments (neurodegenerative diseases or encephalitis) and severe long-term effects, and could in turn orient patient care early upon admission [64,65,66]. Neurodegeneration and/or the modulation of neurodegenerative mechanisms are hallmarks of numerous viral brain infections [27]. Indeed, it is now accepted that viral agents could be considered risk factors in the etiology of some neurodegenerative diseases [27,67,68]. We detected the presence of several neuronal biomarkers involved in neuroinflammation (MIF), neuro-injury (NCAM-1), and neurodegeneration (KLK6, TDP43, and YKL40) in the serum of patients with WNND. The relation of these biomarkers with neuronal diseases, cognitive impairments, or dementia could make them robust targets for the long-term follow-up of WNV patients.

The brain neurovasculature is emerging as a key structure required for brain homeostasis and is often perturbed (i.e., by viruses) and/or has a role in the perturbations underlying neurodegenerative diseases [69,70]. To monitor whether neuroinflammation exists and/or the BBB integrity is affected, the analysis of specific neurobiomarkers in serum, such as Tau, NFL, or GFAP, is now commonly investigated. In this context, we investigated neurotrophic factors in the serum of WNV patients to assess potential brain perturbations occurring during infection. Interestingly, we found significant increases in several factors in the serum of WNND patients, such as TDP-43. TDP-43 is a DNA-binding factor that plays key functions in RNA processing by interacting with ribonucleoproteins and regulating gene expression. Mutations in TDP-43, as well as TDP-43 aggregates, are now considered to have a role in neurodegenerative disorders, such as ALS, AD, and frontotemporal dementia [71,72,73,74]. TDP-43 aggregates have been found to accumulate during Coxsackievirus B3 brain infection, possibly due to a direct effect of viral proteins on the redistribution and cleavage of TDP-43; a process favoring viral replication [75]. Therefore, the major release of TDP-43 in WNV infection could be the result of dysregulation of TDP-43 expression and/or activity, and could thus be considered a potential risk factor for the development of neurodegenerative disease.

YKL40 is a glycoprotein that, among other functions, regulates apoptosis and inflammasomes. YKL40 can be secreted by activated microglia during neuroinflammation [76]. Interestingly, this inflammatory mediator has a role in AD and is now considered a potential biomarker for the differential diagnosis of AD [77,78]. The high serum concentration of YKL40 in our study indicates the monitoring of this protein as a potential biomarker for WNV infection. In addition, YKL40 has been put forward as a potential biomarker for other viral infections, such as SARS-CoV2 and tick-borne encephalitis virus (TBEV) [79,80]. Regarding TBEV infection, differences in YKL-40 levels in CSF samples between patients with meningoencephalitis versus those with meningitis were proposed as a biomarker for the differentiation between clinical forms of TBE [79]. However, here we could not detect significant differences in YKL-40 levels between encephalitis and meningitis in WNV infection. GFAP is a neuronal marker of astrocyte activation, a signature of neuroinvasive diseases including AD [19,81,82], but also during viral brain infections [83,84]. In the serum, GFAP level can be correlated with vascular leakage. In this cohort, the serum concentration of GFAP was elevated in WNV encephalitis and meningitis patient groups, highlighting potential neuronal inflammation and vascular damage, and correlating with a previous WNV infection description [47]. Kallikrein-6 (KLK6) is an enzyme with a major role in neuroinflammatory and neurodegenerative diseases [85]. KLK6 has been considered an early biomarker in the context of traumatic brain injury in rats [86], whereas it has been put forward as a therapeutic target in multiple sclerosis [87] and for AD patient care [88]. The roles of KLK6 in varicella-zoster virus pathogenesis were also examined [89]. KLK6 was proposed as a biomarker for human papilloma virus-positive cervical cancer [90], axonal injury in HIV infection [91], and for patient prognosis in glioblastoma follow-up [92]. In our study, similarly for GFAP, the concentration of KLK6 was found upregulated in sera from WNND patients. This biomarker is further evidence of neuronal damage that should be explored in the context of neuroinfection scoring.

We also report a strong increase in serum BDNF concentration, a key neurotrophic factor involved in the maintenance of CNS cells. The serum concentration of BDNF was higher in WNND patients than in healthy individuals. BDNF is described as elevated in other brain disorders, such as viral encephalitis and bacterial CNS infections [93,94]. The release of BDNF seems to have a role in the severity of CNS injury and its capacity to predict neuronal involvement is increasingly studied [95,96]. Moreover, BDNF is associated with various degenerative diseases and is shown to be upregulated in chronic stress, including ischemia injury or schizophrenia [97,98].

Another interesting marker in our study is GDNF, a neurotrophic factor also involved in neural cell survival. Previously, this molecule was assessed in the context of bacterial or viral encephalitis or meningitis with unconvincing results [99]. Then, a link between the CSF concentration of this marker and HIV brain disease was suggested [100]. Here, we found a significantly elevated level of GDNF in the CSF of WNV meningitis patients when compared to encephalitis patients. The role of GDNF as a biomarker of WNND requires further investigation. Interestingly, we also detected a significant increase in IFNγ and Neurogranin in the CFS of encephalitis patients. Neurogranin is a small protein primarily expressed in granule-like structures in pyramidal cells of the hippocampus and cortex and is involved in synaptic plasticity, synaptic regeneration, and long-term potentiation mediated by the calcium- and calmodulin-signaling pathways [101]. This protein was recently described as a potential biomarker of neurological disorders given its role in several brain diseases, such as AD, Parkinson’s disease, Creutzfeldt–Jakob disease, and neurological complications of HIV, and is expressed in the serum after brain trauma [101,102]. Furthermore, in the context of leukocyte recruitment initiating neuroinflammation, MIF is an important chemokine that is correlated to glial activation and chronic inflammation [103], as well as with BBB impairment [20]. The increased concentration of MIF found in WNND sera could initiate neuroinflammation and BBB injury. Finally, NCAM-1, an adhesion molecule involved in neural cell adhesion and synaptic plasticity, has also been related to neuroinflammation. The expression of NCAM-1 has been found dysregulated during the course of COVID-19 neuroinfection and a potential role in Guillain-Barre syndrome has been reported [104,105]. In our study, increased serum NCAM-1 concentration in WNND patients suggests a peripheral neuronal effect that can occur during WNV infection.

## 5. Conclusions

In this study, we demonstrate systemic inflammation in WNND patients by the release of neuronal molecules related to neuronal diseases and potential neurodegeneration. In addition, this (neuro-) inflammatory condition can prolong for several weeks and could impair BBB integrity. The specificity of and mechanistic implication of these biomarkers in WNV infection and BBB impairment requires deeper evaluation. Altogether, these results give insight into the crucial role of biomarkers and patient monitoring in improving WNV infection care and preventing undesirable outcomes.

## Figures and Tables

**Figure 1 viruses-14-00756-f001:**
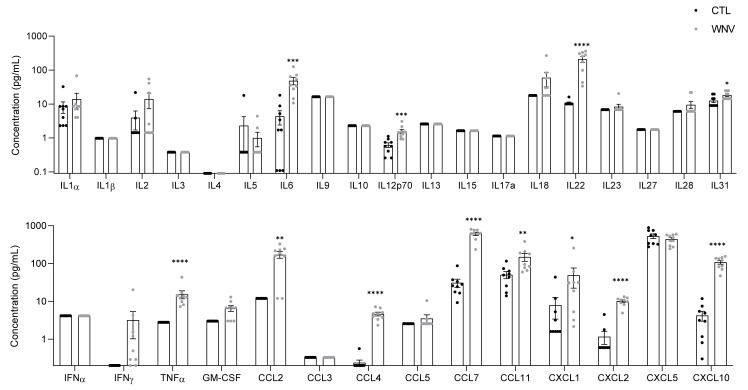
Upregulation of pro-inflammatory mediators after WNV infection in the serum of C57/BL6 mice (6 dpi). Plasma concentration in pg/mL of pro-inflammatory markers measured by multiplex bead-based ELISA in non-infected (black) and WNV-infected mice (grey). Bars show means ± SEM (* *p* < 0.05, ** *p* < 0.01, *** *p* < 0.001, **** *p* < 0.0001).

**Figure 2 viruses-14-00756-f002:**
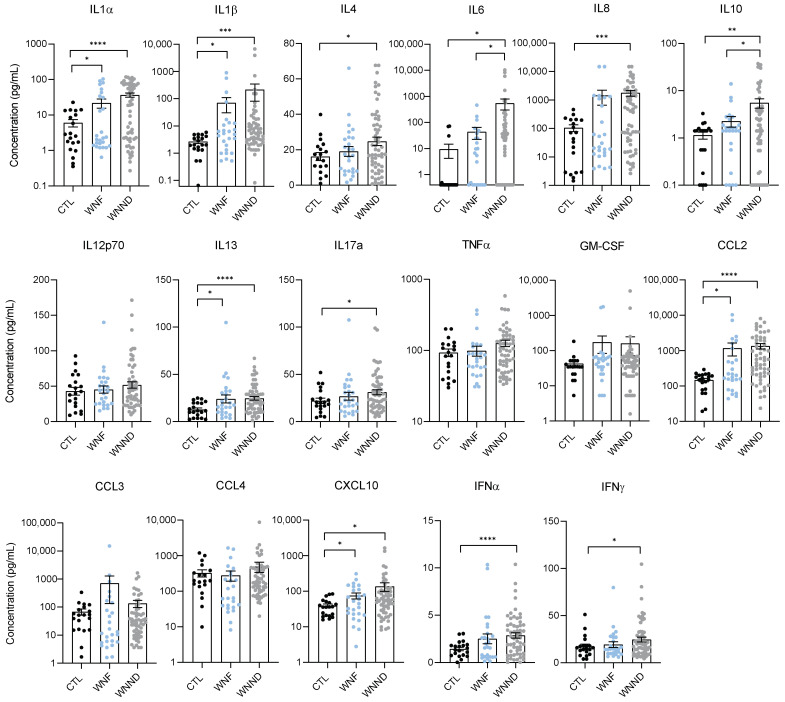
WNV infection results in systemic inflammation in patient sera. Plasma concentration in pg/mL of different pro-inflammatory markers of non-infected CTL (black), WNF (blue) and WNND (grey) humans, measured by multiplex bead-based ELISA. Bars show means ± SEM (* *p* < 0.05, ** *p* < 0.01, *** *p* < 0.001, **** *p* < 0.0001).

**Figure 3 viruses-14-00756-f003:**
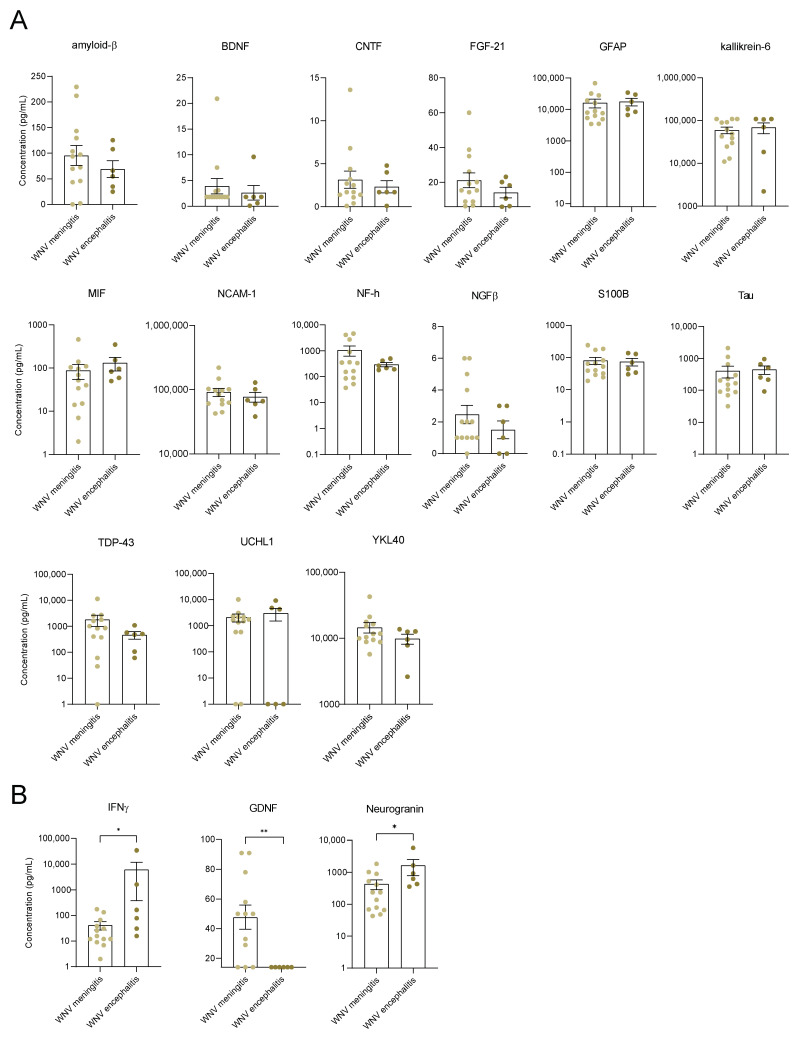
Identification of biomarkers of neuroinflammation in CSF of WNND patients. (**A**) CSF concentration in pg/mL of different neuronal markers of non-infected CTL (black), WNV-meningitis (yellow), and WNV-encephalitis (orange) humans, measured by multiplex bead-based ELISA. (**B**) IFNγ, GDNF, and neurogranin are differentially expressed in patients with meningitis and encephalitis. Bars show means ± SEM (* *p* < 0.05, ** *p* < 0.01).

**Figure 4 viruses-14-00756-f004:**
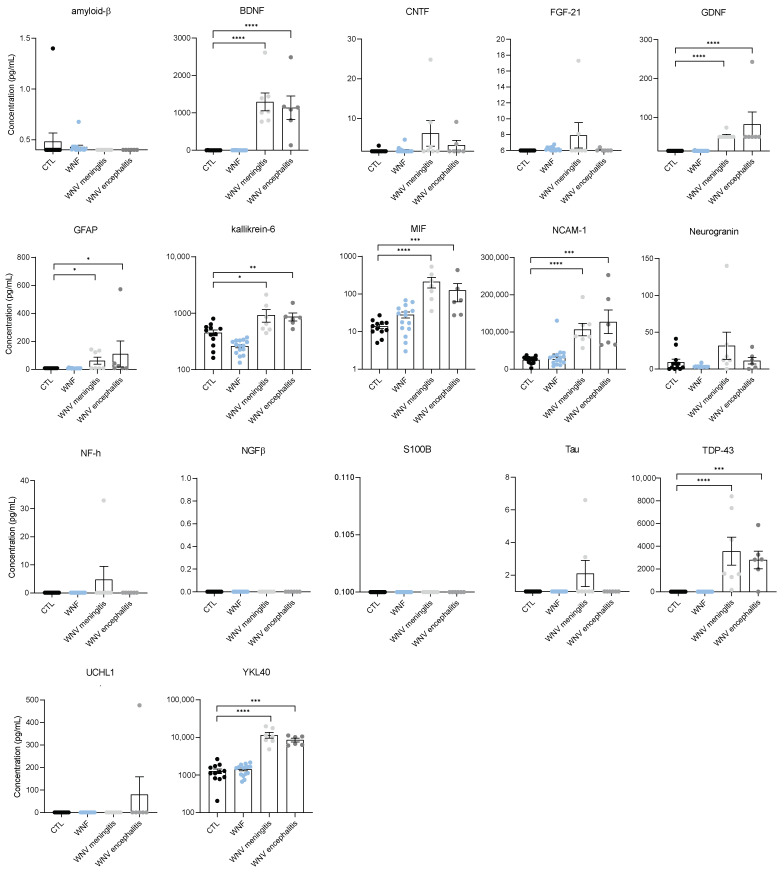
Sera of WNND patients display markers of neuro-inflammation and neuronal injuries. Plasma concentration in pg/mL of different pro-inflammatory markers of non-infected CTL (black), WNF (blue), WNV-meningitis (light grey) and WNV-encephalitis (dark grey) humans, measured by multiplex bead-based ELISA. Bars show means ± SEM (* *p* < 0.05, ** *p* < 0.01, *** *p* < 0.001, **** *p* < 0.0001).

**Figure 5 viruses-14-00756-f005:**
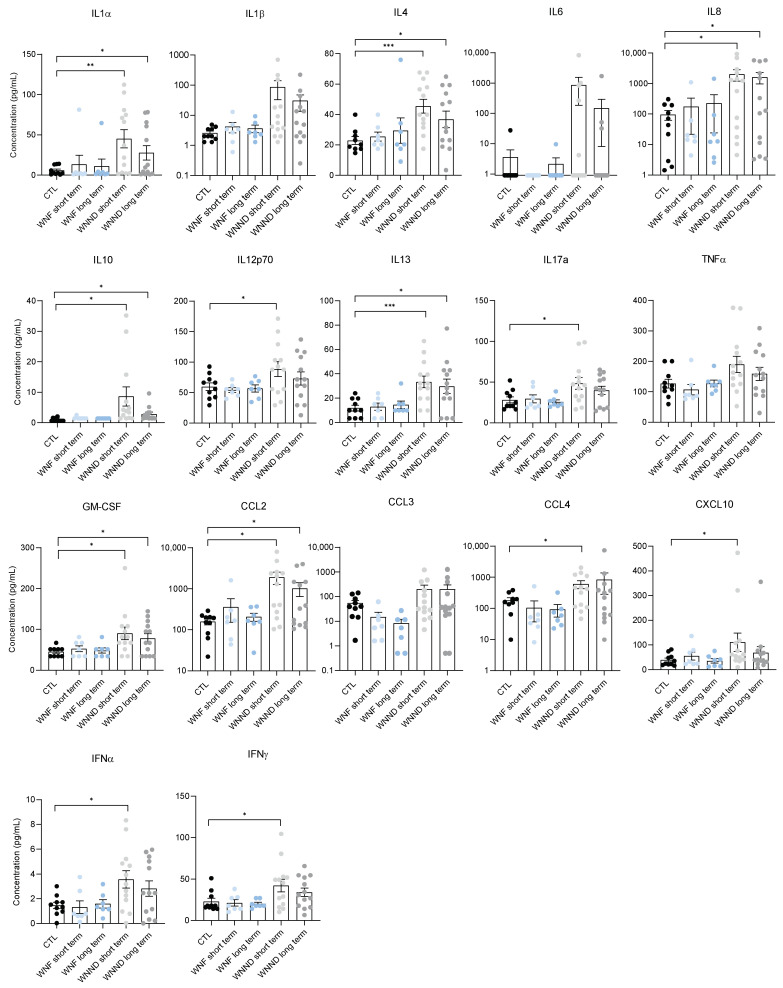
Persistence of inflammation in WNND patient sera. Plasma concentration in pg/mL of different pro-inflammatory markers of non-infected CTL (black), WNVF-short term (light blue), WNVF-long term (dark blue), WNVND-short term (light grey) and WNVND-long term (dark grey) humans, measured by multiplex bead-based ELISA. Bars show means ± SEM ((short term = 2 to 11 days, long term = more than 20 days); * *p* < 0.05 ** *p* < 0.01 *** *p* < 0.001).

**Figure 6 viruses-14-00756-f006:**
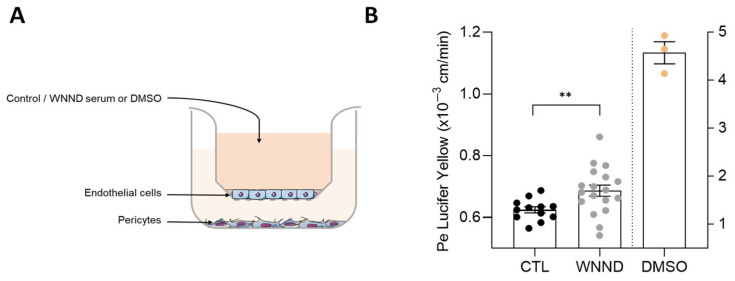
WNND serum slightly perturbs the endothelium in a human BBB model. (**A**) Primary brain pericytes cultured in the basolateral compartment allow differentiation of CD34+ blood cord-derived endothelial cells in human brain-like endothelial cells cultured on transwell filters in the apical compartment. Serum can be added in the apical compartment of this human in vitro BBB model representing the blood. (**B**) Permeability coefficient (Pe) of the paracellular marker Lucifer Yellow was measured 4 days after the addition of CTL or WNND serum or DMSO as a positive control. ** *p* < 0.01.

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
