# Peer review of "West Nile Virus Neuroinfection in Humans: Peripheral Biomarkers of Neuroinflammation and Neuronal Damage"

_viruses, 2022, doi:10.3390/v14040756_

Round 1

Reviewer 1 Report

I find the manuscript interesting and worth publishing. However, it seems a bit lengthy and I think a reader would profit from size reduciton of the paper. 

Author Response

I find the manuscript interesting and worth publishing. However, it seems a bit lengthy and I think a reader would profit from size reduciton of the paper. 

We thank the reviewer for his positive comments. We have reduced the length of the paper by deleting the results on IFNAR mice to refocus the study on the results on humans sample.

Reviewer 2 Report

Manuscript Number: viruses-1643741

West Nile virus neuro infections in humans: peripheral biomarkers of neuroinflammation and neuronal damage.

Constant and colleagues reported characterization of inflammation and neuronal biomarkers released during WNF infection, mainly in the context of neuronal impairment. Identifying those biomarkers might play an important role in patient monitoring to improve care and prevent undesirable outcomes. The paper is well written, and the introduction and the material, and the methods are adequately described. But there are some issues with the experimental approach.

1.Fig1. The experimental outcome is already predetermined in the way the mice are chosen. It is known that the risk factors of developing WNND during WNV infection include old age and immune dysfunction. Therefore, it is expected that 10 times and sometimes even more upregulation of proinflammatory mediators in the serum of INNAR mice compare to C57/BL6. It is also not clear whether the upregulated cytokines are WNV specific Additionally, the kinetics of the early stage of viral replication and the severity of the early symptoms of WNV infection predetermine the level of cytokines and chemokines in serum. The authors need more explanation in the discussion part for the difference they observed in this experimental setting.

2.Fig2. Here the authors showed the difference in the systemic inflammatory profile between patients with WNF versus WNND. They found high pro-inflammatory markers in WNND patients. The study by Weatherhead et al 2015 showed that 27% of patients diagnosed with WNF without being diagnosed WNND had neurological abnormalities up to three years post-infection. They also reported that such neurological abnormalities appear to occur equally in patients diagnosed with WNND compare with patients diagnosed with WNF. This begs the question of how exactly the patients with WNND and WNF are diagnosed and whether there is a direct association between the development of WNND with development of neurological abnormalities. This brings the equation of how you see such a difference in your experimental approach 

Fig3 and4.

Unexpectedly, the authors did not see much difference between WNV meningitis and WNV encephalitis patients' neuroinflammation given the fact that those patients with WNE show complete recovery and while those with WNE have a greater risk of long term sequelae.

4. please correct the sentence on page 11 line 249. delete one of the infinitives (to decrease)

Author Response

West Nile virus neuro infections in humans: peripheral biomarkers of neuroinflammation and neuronal damage.

Constant and colleagues reported characterization of inflammation and neuronal biomarkers released during WNF infection, mainly in the context of neuronal impairment. Identifying those biomarkers might play an important role in patient monitoring to improve care and prevent undesirable outcomes. The paper is well written, and the introduction and the material, and the methods are adequately described. But there are some issues with the experimental approach.

We thank the reviewer for his interest and useful recommendations.

1.Fig1. The experimental outcome is already predetermined in the way the mice are chosen. It is known that the risk factors of developing WNND during WNV infection include old age and immune dysfunction. Therefore, it is expected that 10 times and sometimes even more upregulation of proinflammatory mediators in the serum of INNAR mice compare to C57/BL6. It is also not clear whether the upregulated cytokines are WNV specific Additionally, the kinetics of the early stage of viral replication and the severity of the early symptoms of WNV infection predetermine the level of cytokines and chemokines in serum. The authors need more explanation in the discussion part for the difference they observed in this experimental setting.

Based on these comments and those of other reviewers, we decided to delete the section on IFNAR mice that seemed to lack clarity. We kept the results on the immunocompetent mice which represent a more homogeneous group of 3-week-old mice.

2.Fig2. Here the authors showed the difference in the systemic inflammatory profile between patients with WNF versus WNND. They found high pro-inflammatory markers in WNND patients. The study by Weatherhead et al 2015 showed that 27% of patients diagnosed with WNF without being diagnosed WNND had neurological abnormalities up to three years post-infection. They also reported that such neurological abnormalities appear to occur equally in patients diagnosed with WNND compare with patients diagnosed with WNF. This begs the question of how exactly the patients with WNND and WNF are diagnosed and whether there is a direct association between the development of WNND with development of neurological abnormalities. This brings the equation of how you see such a difference in your experimental approach 

In our study the categorization of ‘WNF’ and ‘WNND’ is always based on the clinician’s determination at hospital admission (and symptoms/information that they share). We have no information regarding the long-term progression of the patient’s condition (i.e. several years after infection). Several factors probably contributed to the late onset of neurological abnormalities in WNF patients, such as age and other underlying diseases. It can also depend on the genetic factors of the host or other infection. Besides the presence of the biomarkers of the acute neuro-infection, patients with two or more comorbidities could be predisposed for developing severe neurological status after hospital admission or long-term impairments as neurological abnormalities. We added additional information on the cohort in the M&M section.

Fig3 and4.

Unexpectedly, the authors did not see much difference between WNV meningitis and WNV encephalitis patients' neuroinflammation given the fact that those patients with WNE show complete recovery and while those with WNE have a greater risk of long term sequelae.

It is true that patients who initially presented with acute WNE have high rates of persistent neurological abnormalities over time compared with those who presented with acute WNF or WNM. Interestingly we did not see a significant difference in the sera and CSF between meningitis and encephalitis except for some markers like GDNF or neurogranin in the CSF. Either these markers play an important role in the evolution of the disease, or the long-term maintenance of other factors could play an important role. It is true that some long-term (>20 days post infection) sera were also analyzed, however the maximum was 125 days between the symptom onset and sampling and long-term serum was not available from each patient. A more complete analysis of different biomarkers over the long term (several years after infection) between WNV meningitis and encephalitis would be usefull to try to answer this question. We have added a section in the discussion to address this point (lines 291-301).

  1. please correct the sentence on page 11 line 249. delete one of the infinitives (to decrease)

We fixed it.

Reviewer 3 Report

In this paper, Constant et al., evaluate the cytokine levels from patients that were infected with West Nile virus during the Hungary outbreak in 2018. Authors compare the levels of cytokines between patients who had mild disease and various forms for severe WNV neurological disease. They evaluate these parameters mostly in the serum and some in CSF. They also perform time-course evaluation (short-term vs long-term) of cytokine levels from these patients. They also use mouse infection to evaluate similar cytokine responses (WT and IfnaR-/-) and drive home the message that severe neurological disease is associated with higher inflammatory cytokine signature. The data is highly interesting since it is acquired from human samples, has good sample size and hence will provide valuable information regarding the heterogeneity of disease associated with WNV infection. However, there are several flaws and proper controls missing in the paper. These need to be addressed and stated clearly in the paper for it to be meaningful. My suggestions and critical points of are stated below.

Major points:

  1. The authors examine cytokine levels in serum at Day 3 for IFNAR-/- mice compared to controls (uninfected). However, they examine Day 6 for WT B6 mice. It is understandable that IFNAR-/- do not survive until Day 6. But for comparison, it will be good to add data for WT B6 mice on Day 3 post infection. There is possibility that some cytokines/ chemokines increase at Day 3 in WT mice and not at Day 6 (for example: In Figure 1A: IFN-alpha levels are same between control and WNV; since IFN-alpha is clearly upregulated in many viral infections, it’s upregulation might be missed by looking at just one time point). Adding day 3 WT B6 data will also be a good way of comparing with IFNAR-/- Line 165-168: it is not appropriate to compare WT mice and IFNAR-/- mice since their serum was isolated on different days (and mice were likely different age? See point 3)

  1. The rationale for using IFNAR-/- mice is not clearly stated in the paper. One assumption would be that authors suggest that IFNAR-/- mice being more susceptible to infection, can be thought as a model of severe infection in humans. This seems very loose connection and does no add much to the paper overall. Nevertheless, it is important for authors to state the reason clearly in the paper. It will also be beneficial to add mouse data as last figure and see if there are any correlations in cytokine levels between humans and mice.

  1. In the method section, authors mention they used 3-week-old WT B6 mice whereas 2 to 6 months old IFNAR-/- Why these different ages of mice? Age is an important factor that contributes to disease severity in humans as well as mice for WNV infection. It will be good to use mice of same age. Or if authors propose to evaluate older mice (since human patients average age in paper is >50) – to that end it will be ideal to use older mice (Reference: Funk and Klein; 2021; Aging Cell)

  1. Authors refer to all cytokines as inflammatory type. But certain cytokines that were evaluated belong to the type 2 immune response such as IL-4 and IL-13. It is important to mention this in paper. Moreover, IL-10 is well known for immuno-regulatory role. It is also good to speculate their role probably in discussion section. Since, only IFNAR-/- mice or WNND patients show upregulation of these particular cytokines, one hypothesis is that these type 2 cytokines or immunoregulatory cytokines are induced once pro-inflammatory cytokines rise above a certain threshold to counteract excessive inflammation (collateral tissue damage).

  1. In Figure 3A, comparing serum samples from control to CSF from patients is misleading and should be omitted. Authors acknowledge the problem in results section, so it is better to omit this control or add this figure as supplemental.

  1. In Figure 4, authors examine cytokine levels and neuroinflammatory markers in serum of patients with WNV meningitis and WNV encephalitis with control group (healthy). What about patients with WNV fever (mild disease/ non-neurological)? This is an ideal group to compare to know the differential in neuroinflammation between mild infection vs. severe (neurological) infection.

  1. Line 248, Figure 5: “IL-10 showing rapid decrease’’ – is this statistically significant? There is no stats asterisk comparing WNND short-term vs long-term.

  1. Figure 6: Does serum from WNVF perturb the endothelium in human BBB model?

  1. Authors show human samples with cytokine levels in serum and CSF – short-term and long-term. But similar analysis is not done in mice. Can authors check if infection of WTB6 mice causes changes in cytokine levels and neuro-inflammatory signatures in serum and CSF over time?

Minor points

  1. The authors refer the mouse model - type I interferon receptor deficient mice as Ifnar-/-. It is important to be specific and clarify whether mice are deficient in IfnaR1-/- or IfnaR2-/- and use this specific terminology throughout the paper when referring to these mice.

  1. The route of infection that is widely used in studying West Nile virus infection of mice is subcutaneous footpad injection, since it closely mimics natural route (mosquito bite). However, authors used intraperitoneal route. What is the rationale for using systemic route of infection rather than footpad? It will be good to mention this in the paper.

  1. Line 159: please correct in results section – data description is for WT mice (according to figure legends) it should be WT mice not IFNAR-/- mice.

  1. What was the status of the patients in terms of comorbidities both WNV and healthy? How do authors define ‘healthy’? Is appropriate way of referring to them ‘WNV negative’? It will be good to add a table for human patients with signs and symptoms of disease, age and sex along with if these patients had some underlying disorders. Did healthy individuals have any underlying disease manifestations and were they negative for other common pathogens? One recently described key factor (at least in mouse studies) is coinfection with parasites that make mice more susceptible to severe infection (Reference: Desai et al; 2021; Cell). Were these patients tested for possible helminth or other parasitic infection? It will be good to add this to discussion paragraph as possible causes of disease variability in patients.

Author Response

Reviewer #3:

In this paper, Constant et al., evaluate the cytokine levels from patients that were infected with West Nile virus during the Hungary outbreak in 2018. Authors compare the levels of cytokines between patients who had mild disease and various forms for severe WNV neurological disease. They evaluate these parameters mostly in the serum and some in CSF. They also perform time-course evaluation (short-term vs long-term) of cytokine levels from these patients. They also use mouse infection to evaluate similar cytokine responses (WT and IfnaR-/-) and drive home the message that severe neurological disease is associated with higher inflammatory cytokine signature. The data is highly interesting since it is acquired from human samples, has good sample size and hence will provide valuable information regarding the heterogeneity of disease associated with WNV infection. However, there are several flaws and proper controls missing in the paper. These need to be addressed and stated clearly in the paper for it to be meaningful. My suggestions and critical points of are stated below.

We thank the reviewer for its thorough analysis of our manuscript and its positive appreciation of our work. Here are our responses to the points raised.

  1. The authors examine cytokine levels in serum at Day 3 for IFNAR-/- mice compared to controls (uninfected). However, they examine Day 6 for WT B6 mice. It is understandable that IFNAR-/- do not survive until Day 6. But for comparison, it will be good to add data for WT B6 mice on Day 3 post infection. There is possibility that some cytokines/ chemokines increase at Day 3 in WT mice and not at Day 6 (for example: In Figure 1A: IFN-alpha levels are same between control and WNV; since IFN-alpha is clearly upregulated in many viral infections, it’s upregulation might be missed by looking at just one time point). Adding day 3 WT B6 data will also be a good way of comparing with IFNAR-/-Line 165-168: it is not appropriate to compare WT mice and IFNAR-/- mice since their serum was isolated on different days (and mice were likely different age? See point 3)
  2. The rationale for using IFNAR-/- mice is not clearly stated in the paper. One assumption would be that authors suggest that IFNAR-/- mice being more susceptible to infection, can be thought as a model of severe infection in humans. This seems very loose connection and does no add much to the paper overall. Nevertheless, it is important for authors to state the reason clearly in the paper. It will also be beneficial to add mouse data as last figure and see if there are any correlations in cytokine levels between humans and mice.
  3. In the method section, authors mention they used 3-week-old WT B6 mice whereas 2 to 6 months old IFNAR-/-Why these different ages of mice? Age is an important factor that contributes to disease severity in humans as well as mice for WNV infection. It will be good to use mice of same age. Or if authors propose to evaluate older mice (since human patients average age in paper is >50) – to that end it will be ideal to use older mice (Reference: Funk and Klein; 2021; Aging Cell).

We initially analyzed IFNAR mice infected or not by WNV and all sacrificed at 3 dpi. We had done the same experiment at 6 dpi on C57/BL6 immunocompetent mice. We did not want to compare these two groups with each other. Based on the reviewer’s comments, we decided to delete the part of the study concerning the IFNAR mice, in order to make the document more clear for readers. We kept the part on immunocompetent mice. WNV-infected mice and CT mice were sacrificed at 6 dpi.

  1. Authors refer to all cytokines as inflammatory type. But certain cytokines that were evaluated belong to the type 2 immune response such as IL-4 and IL-13. It is important to mention this in paper. Moreover, IL-10 is well known for immuno-regulatory role. It is also good to speculate their role probably in discussion section. Since, only IFNAR-/- mice or WNND patients show upregulation of these particular cytokines, one hypothesis is that these type 2 cytokines or immunoregulatory cytokines are induced once pro-inflammatory cytokines rise above a certain threshold to counteract excessive inflammation (collateral tissue damage).

We added a part in the discussion concerning this important point (lines 290-294).

In Figure 3A, comparing serum samples from control to CSF from patients is misleading and should be omitted. Authors acknowledge the problem in results section, so it is better to omit this control or add this figure as supplemental.

We have removed this « control » from the figure. 

  1. In Figure 4, authors examine cytokine levels and neuroinflammatory markers in serum of patients with WNV meningitis and WNV encephalitis with control group (healthy). What about patients with WNV fever (mild disease/ non-neurological)? This is an ideal group to compare to know the differential in neuroinflammation between mild infection vs. severe (neurological) infection.

We investigated systemic inflammation in WNND versus WNF patients in the first part of our study. We wished to focus the second part of our work on the study of neuronal markers by starting their comparative study in patients with meningitis vs encephalitis in CSF and then in serum. Because CSF is only available for neurological forms of the disease (WNND) we did not perform this part of the study on WNF patients. Nevertheless the reviewer is right in indicating that a comparison of the expression of these markers with the WNF serum is of interest. We performed this analysis and we did not find any increase of these markers in WNF patients unlike WNND patients. We included these results in the corresponding figure.

  1. Line 248, Figure 5: “IL-10 showing rapid decrease’’ – is this statistically significant? There is no stats asterisk comparing WNND short-term vs long-term. 

As this data is not statistically significant we have removed it from the paper

  1. Figure 6: Does serum from WNVF perturb the endothelium in human BBB model?

As mentioned previously, the second part of the study focused mainly on WNND sera in which we observed inflammation. For this reason we did not investigate the variation in endothelial permeability in WNVF sera. We are unable to repeat these experiments due to the difficulty in setting up the cell model used and the amount of human serum required to have statistically significant data. 

  1. Authors show human samples with cytokine levels in serum and CSF – short-term and long-term. But similar analysis is not done in mice. Can authors check if infection of WTB6 mice causes changes in cytokine levels and neuro-inflammatory signatures in serum and CSF over time?

Since the mice die only after a few days after infection in our experiments, we were unable to perform time-based analyses on infected mice.

Minor points

1.The authors refer the mouse model - type I interferon receptor deficient mice as Ifnar-/-. It is important to be specific and clarify whether mice are deficient in IfnaR1-/- or IfnaR2-/- and use this specific terminology throughout the paper when referring to these mice.

3.Line 159: please correct in results section – data description is for WT mice (according to figure legends) it should be WT mice not IFNAR-/- mice.

The section on IFNAR mice has been removed.

2.The route of infection that is widely used in studying West Nile virus infection of mice is subcutaneous footpad injection, since it closely mimics natural route (mosquito bite). However, authors used intraperitoneal route. What is the rationale for using systemic route of infection rather than footpad? It will be good to mention this in the paper.

In the laboratory we compared ip and sb injections without observing major differences in systemic cytokine production. Nevertheless, the C57/BL6 mice were injected subcutaneously for this specific study, we corrected this mistake.

  1. What was the status of the patients in terms of comorbidities both WNV and healthy? How do authors define ‘healthy’? Is appropriate way of referring to them ‘WNV negative’? It will be good to add a table for human patients with signs and symptoms of disease, age and sex along with if these patients had some underlying disorders. Did healthy individuals have any underlying disease manifestations and were they negative for other common pathogens? One recently described key factor (at least in mouse studies) is coinfection with parasites that make mice more susceptible to severe infection (Reference: Desai et al; 2021; Cell). Were these patients tested for possible helminth or other parasitic infection? It will be good to add this to discussion paragraph as possible causes of disease variability in patients.

We added more information in the M&M section about the healthy patients and the cohort of WNV patients (age, gender, symptoms …). The healthy patients were from occupational medicine and had no signs of disease or infection (negative for WNV and for common pathogens). In our introductory part of the discussion, on inter-patient variability, we added a reference mentionning the influence of parasitic co-infections. We do not have information on possible parasitic infections for the patients of this cohort.

Round 2

Reviewer 3 Report

The changes made by the authors, new additions/deletions to the text and their response to reviews is satisfactory.